# Students' Perceptions of an EFL Vocabulary Learning Mobile Application

**Blanka Klimova** [1,*] and **Petra Polakova** [2]

1   Department of Linguistics, University of Hradec Kralove, Rokitanskeho 62/26, 500 03 Hradec Kralove, Czech Republic

2   Department of Language Pedagogy and Intercultural Studies, Constantine the Philosopher University, 949 74 Nitra, Slovakia; petra.polakova@ukf.sk

*   Correspondence: blanka.klimova@uhk.cz

**Abstract:** Mobile devices have penetrated all spheres of human activities, including education. Previous research has shown that smartphones are becoming widely used in learning as they can improve knowledge retention and increase student engagement. The purpose of this study was to discuss students' perception of the use of a mobile application aimed at learning new English vocabulary and phrases and describe its strengths and weaknesses as perceived by the students. In total, 28 university students answered a pen and paper questionnaire survey after experiencing the app during one semester. Overall, the students' agreement to the positive aspects displayed in the questionnaire prevailed over their disagreement or neutral opinions. The mobile app helped students prepare for the final achievement test, learning was accessible from anywhere and at any time, students appreciated the corrective feedback and would opt for the implementation of the mobile app in other courses taught at the faculty. On the other hand, as the findings indicate, the students reported that the app was not very supportive regarding communication performance; they did not find the teachers' notifications encouraging and they did not use the pronunciation support much, which was caused by various factors, such as offering students words and phrases without context or not testing all the items in the final credit test. The findings of this study contribute to the existing knowledge of students' perceptions of the use of mobile apps for learning purposes.

**Keywords:** smartphones; mobile applications; EFL; university students; attitudes

## 1. Introduction

Today, young people cannot imagine their life without smartphones. University students usually have their smartphones at hand even during their classes. Nearly every student owns one or two smartphones [1]. As statistics [2] point out, 64% of students consider accessing their learning material from a mobile device essential. Smartphone learners complete course material 45% faster than those using a computer and 89% of smartphone users download applications, 50% of which are used for learning.

Therefore, teachers try to exploit these mobile devices in their teaching too. This is also true for teaching English as a foreign language (EFL). Nowadays, mobile-assisted language learning (MALL) appears to be well established EFL methodology, especially at a university level [3–5] because it enables students to carry this mobile device easily and study from it anywhere, at any time, and according to their own pace [6]. Moreover, students can get immediate feedback on their performance [3,7]. Apart from the immediate feedback, perceived content is an important factor that affects user satisfaction with a mobile application. According to the findings by Liu et al. [8], the knowledge content should be clear, concise and refined to reflect the smartphones interface features, as well as to provide effective

learning experiences. In addition, Tayan [9], in his research study, reported that MALL facilitates autonomous learning and better collaboration in a richer learning environment.

Research also shows that students are more motivated to learn when using mobile devices [9–12]. For instance, the findings by Tayan [9] showed that 85% of university students and 100% of teachers thought that mobile learning could act as a motivational stimulus for learning a foreign language. This is also connected with the fact that students generally have a positive attitude towards the implementation of mobile devices into foreign language classes [13,14].

As a consequence, by using mobile devices in EFL, students may achieve better academic results in comparison with those not using mobile devices, as has been shown in research studies [15–17]. Ozer and Kilic [16] conducted an experiment with 63 university students studying an English language preparatory course. Their level of English was A2 according to the Common European Reference Framework for Languages [18]. All students practiced all four language skills. The intervention period lasted six weeks and they had 32 lessons of English. The experimental group consisting of 32 students using mobile applications significantly outperformed the control group (31 students) not using the mobile apps. The findings also revealed that students' results did not differ as far as the gender and field of study were concerned. Similar positive findings were confirmed by Klimova and Prazak [15] in their study on the evaluation of the effectiveness of the use of a mobile application for learning new vocabulary on students' study achievements. They found that students using the mobile app had significantly higher learning outcomes than students who did not use this app. Furthermore, Hwang et al. [17], in their study, proposed that game-based learning activities may significantly improve students' speaking skills if driven by a mobile device, especially in enhancing the practice of speaking English as a foreign language, the development of meaningful sentences and speaking with greater accuracy and confidence.

At present, mobile applications are the most common applications in the English m-learning context [19]. They are used for the teaching and learning of all four language skills (i.e., speaking, writing, listening, and reading) [16,20–23], as well as vocabulary [7,15,24–26], pronunciation [27], and grammar structures [28]. Nevertheless, as Heil and et al. [29], in their study, pointed out, most of the mobile applications focus on vocabulary learning. In their evaluation of 50 top commercial apps aimed at language learning, 84% of these apps concentrated on practicing vocabulary items as isolated units and 53% assessed vocabulary in context, which is important for enhancing students' vocabulary knowledge through incidental repeated exposure to new words.

Nowadays, these mobile applications for EFL learning are exploited as part of the so-called blended learning approach to teaching EFL. Poláková [30] sees blended learning as an approach to education that combines online interaction with traditional classroom methods. As suggested by Polakova [30], combining face-to-face practices with technology-based activities may make the learning process more interesting and help students to increase their motivation. Authors such as Mikulecky [31] also claimed that blended learning is one of the modern and promising approaches to learning. It aims to integrate traditional learning with innovative methods of learning in order to create an effective learning environment aiming to enrich learning experience. As claimed in Khan [32], the charm of the blended learning approach lies in the connection of technology aided learning methods with traditional based learning. According to him, a valuable learning process is created by employing various learning methods, tools and techniques. He believes that there are several advantages of blended learning in comparison with using only traditional, face-to-face teaching. Firstly, using mobile devices, learners are allowed to access the learning materials on their own, at anytime, anywhere and on their own pace. Secondly, blended learning is considered to be more effective since it combines more approaches. Thirdly, using mobile technology within blended learning also spares the anxiety and embarrassment since learners can ask their teachers any question through mobile devices. Learning through mobile technology ensures communication between peers and teachers, and enables quick feedback, too. This environment provides several resources of learning which enhance learners´ confidence and competency [32,33].

There are also some drawbacks of the use of mobile apps for learning, such as technical aspects of the mobile devices (i.e. small screen and keypad) or the Internet does not have to be available everywhere. Students may be distracted by incoming calls or messages. Sometimes, informal learning outside the classroom might be affected by outdoor noise [34].

With mobile devices in the classroom, the teachers play a more active role and they need to become designers of learning experiences for their students [35]. Even if students are proficient users of their personal devices, studies show that without guidance, they may struggle to use them effectively. One of the roles of the teacher is to lead the students to see the potential of mobile learning and to see the potential of becoming a personally empowered learner with the possibility of becoming part of an online community. This can be achieved in several ways, such as using classroom time to allow students to discuss their learning experience outside the school, enabling students to ask questions related to their learning online, or encouraging students to seek out and recommend online resources they find useful. This approach could lead to discussions on the advantages of different apps or services, potentially increasing the motivation of the students. In a classroom in which students are fully engaged in the learning process, there are unlikely to be bored or distracted students. Introducing mobile devices as a new teaching and learning tool follows an instructional pattern where the teacher is a guide and facilitator [36].

Based on the previous statements, learning languages can be supported by using mobile technology; however, it is up to teacher to become a role model, to teach how to incorporate mobile applications in the learning routines, and to raise students´ motivation to use mobile devices as learning tools in the classroom and beyond the classroom environment in order to develop their language proficiency. Therefore, s/he has to respond to students' language and social needs to achieve this.

The purpose of this study was to discuss students' perceptions of the use of a mobile application, which was developed on the basis of students' needs, and describe its strengths and weaknesses in order to make EFL vocabulary teaching and learning effective.

## 2. Methods

### 2.1. Methodology

A pen and paper questionnaire was completed by 28 Czech students (21 females and 7 males) of Management of Tourism in their third academic year. It aimed at exploring students' perceptions about the use of a mobile app which they were using during the winter semester of 2019 for learning new words and phrases. The app was tailored to their needs, especially to their study interests and the skills they wanted to practice. The questionnaire consisted of 13 statements and two open-ended questions modified according to [37]. The questionnaire and its findings are presented in the Results and Discussion section below. A five-item scale ranging from "strongly disagree" to "strongly agree" was used.

### 2.2. Study Design

The mobile app was used by students of Management of Tourism in their third year of their study at the Faculty of Informatics and Management in Hradec Kralove as additional support for learning English in the winter semester of 2019. The semester lasted 13 weeks and the mobile app was used outside the contact English classes. All the contact classes lasted 90 minutes per week, were held from the end of September to mid-December, and focused on the development of all language skills, i.e., listening, reading, speaking, and writing. All 28 students decided to use the mobile app in addition to their contact classes and the learning materials from their English classes stored in a supporting online course. Students' level of English was between B2 and C1 according to the Common European Reference Framework for Languages [18].

The content of the app was aimed at practicing and retaining new words and phrases, which proved to be students' main weakness when learning English, as based on their needs analysis carried

out at the beginning of the semester. The mobile app is called Angličtina (English) TODAY. It was devised by a PhD computer science student in cooperation with the language teacher and so far, it has been only used by the students of Management of Tourism. It consists of the server part, the web application and the mobile application [38]. The mobile app was developed both for the Android and iOS operating systems. There are ten lessons of vocabulary and 10 lessons of phrases. The students must translate the word or the phrase from their native language into English. Each vocabulary lesson is done as a test and comprises of 15–18 new words on average. The same is true for the phrases lessons, which include 10 new phrases on average. In addition, students received notifications sent to them on their smartphones at least twice a week by their teacher in order to remind them to study on a regular basis.

## 3. Results and Discussion

The findings of the pen and paper questionnaire are presented in Table 1 below. It was filled in by 28 students. All the students were used to implementing smartphones for educational purposes. Their answers to the statements are provided in percentages. The replies to the two open questions are discussed below. The authors interpreted the 'agree' and 'strongly agree' responses as students positively valuing the learning statement proposed in the item questionnaire. The 'strongly disagree' and 'disagree' responses were interpreted as students valuing the statements as negative. The items in which positive answers sum more than 51% are interpreted. The rest of the answers were thus neutral.

**Table 1.** Questionnaire findings on students' perceptions about the use of the EFL vocabulary learning app.

| | Strongly Disagree | Disagree | Neutral | Agree | Strongly Agree |
|---|---|---|---|---|---|
| I enjoyed using a mobile app to learn. | 7% | 14% | 46% | 22% | 11% |
| Using the app helped me become more confident in my learning. | 7% | 29% | 32% | 25% | 7% |
| The app was more accessible than books when I was moving around. | 7% | 11% | 11% | 39% | 32% |
| The app had a positive effect on my study behavior. | 3% | 25% | 57% | 7% | 7% |
| The app gave me confidence knowing I had my resources at hand and could access it at any time. | 7% | 7% | 39% | 33% | 14% |
| I checked the pronunciation of the words I was learning on the app. | 7% | 36% | 18% | 32% | 7% |
| Interacting with the app helped me remember my English vocabulary better. | 0% | 11% | 39% | 36% | 14% |
| I appreciated the corrective feedback of the app. | 3% | 14% | 25% | 43% | 14% |
| The notifications sent by the teacher helped me study regularly. | 11% | 25% | 28% | 25% | 11% |
| Using a mobile app to test my vocabulary knowledge was more fun and less stressful. | 7% | 7% | 40% | 39% | 7% |
| The app helped me prepare for the final test. | 3% | 14% | 29% | 32% | 22% |
| Using the app helped me enhance my communication performance. | 11% | 29% | 46% | 14% | 0% |
| I would like the app to be implemented in future courses. | 7% | 14% | 25% | 40% | 14% |

What did you enjoy most about the app? What did you enjoy least about the app?

On the basis of the authors' interpretation of the results stated above, four statements were found to be positive, as more than 51% of the responses were rated as 'agree' or strongly agree'. Firstly, it is clear that students appreciated the accessibility of the mobile app anywhere and at any time more than the traditional books, provided there was Internet access. This is in line with the findings of a Malaysian study [14] whose authors surveyed 33 students on the acceptance of mobile app for language learning. They found that it was easy for the students to download the app, which is easy to carry everywhere and accessible at their preferred time and place. Secondly, students also liked the corrective feedback provided through the mobile app. This is in line with the findings of other research studies [7,8]. It is also important to note that the corrective feedback could be understood as a formative assessment here since the vocabulary and phrases were assessed on a weekly basis, usually twice a week, which was encouraged by teachers' notifications sent to students on their smartphones. In fact, the teacher's aim was to enhance the acquisition of individual words and phrases, which would consequently contribute to a better communicative performance in a foreign language. For example, Das et al. [39] stated that 78% of their students reported that the feedback collected from formative assessment was important for them because it helped to reduce their learning gaps. As Febriani and Abdullah [40] maintain, the implementation of formative assessment in a blended learning environment is rising as it can enhance the learning quality. The findings of other research studies also show that formative assessment has a significant impact on higher academic achievement levels [41,42]. Thirdly, this evidences the third positive answer, which states that the app helped the students to prepare for the final test that was passed by all students except one. Finally, students would welcome the app to be implemented in future courses. This might indicate one of two facts: either that students considered the mobile app as a common and inseparable learning tool, or that they really liked this new way of learning via the mobile app.

In addition, the findings reveal that half of the students agreed that they had remembered the new words and phrases better when interacting with the app [7,15,37]. This was also evidenced by Wu [43] who reported that her university students had remembered around 89 more new English words than the students who had not used the smartphone application. In particular, interactivity, as well as the use of different multimedia, play an important role in this respect [19,37]. Deris and Shukor [14] also emphasized, in their study, that it is particularly the game which stimulates students' to learn vocabulary.

In total, 46% of the students also enjoyed using the mobile app when learning new words and phrases. This was also one of the most frequent replies in their open-ended questions. They reported that it had been more fun, less stressful, easy to use, and a better way to learn than from the textbooks. This is in line with a research study by Kwangsawad [13] and Elaish et al. [19], who reported that their students also had found using smartphones in EFL classes more fun, beneficial and productive. Similar findings were provided by Deris and Shukor [14]. As Ozer and Kilic [16] maintain, the more students use the mobile devices, the more academic gains they are able to achieve.

Although students enjoyed the correction feedback of their performance, they considered it to be very strict, even if they made just a small mistake, such as the lack of a full stop, as it was revealed on the basis of the information provided from the two open questions. Nevertheless, this was done on purpose and should make students realize how important accuracy is, for example, in developing EFL writing. Another feature that was not used by the majority of the students was audio pronunciation of the new words. This might be connected with the fact that these students were already quite confident users of English (i.e., their level of English ranged between B2 and C1) and did not need to check the pronunciation of the new words. Another reason also might be the fact that pronunciation was not part of the test, which was done in a written way only. Students also did not become more confident in their learning by using the mobile app and they did not feel that the app helped them to enhance their communication performance. This might be connected with the fact that the app focused rather on the assessment of individual words and phrases and not the whole complex utterances, which then were developed in the face-to-face class. In fact, this is probably the biggest drawback of the mobile

apps since the technical aspects limit the ability to develop complex utterance speech on one screen. However, bottom-up vocabulary processing can also be an optimal learning strategy, especially if it is associated to a later stage in which the learner can use it in context, as it is the case of the presented blended learning procedure here (cf. [44]). Therefore, at this age, being outside the inner circle of native speakers, they have to consciously drill the new vocabulary and the mobile app can help in this respect. Finally, there was no consensus on the notifications described above. Half of the students appreciated this and reported that the notifications helped them study regularly and the other half did not, which was then also reflected in the fact that most of the students replied that the app had had a neutral effect on their study behavior. Again, this might have been caused by receiving the notifications in a not suitable time of the day. The limitations of this article consist in a small sample of the respondents, as well as in the use of only one survey method. However, the results comply with the findings of other research studies on this topic [3,6,7,13,14,16,37].

## 4. Conclusions

The findings of this study indicate that students perceived the mobile app as facilitative for some learning actions, such as its accessibility from anywhere and at any time, its corrective feedback, and it offering students another opportunity to prepare for the final credit test. In addition, students would opt for the implementation of the mobile app in other courses taught at the faculty. On the other hand, as the findings indicate, the students reported that the app was not very supportive regarding communication performance; they did not find the teachers' notifications encouraging and they did not use the pronunciation support much, which was caused by various factors, such as offering students words and phrases without context or not testing all the items in the final credit test. Therefore, the teachers should always think about the purpose of the use of the mobile app in promoting students' learning in order to generate higher learning outcomes [45].

The findings of this study contribute to the existing knowledge of students' perceptions of the use of mobile apps for learning purposes. However, more research studies should be conducted on students' attitudes toward the use of EFL mobile apps to discover what other aspects might enhance their learning.

**Author Contributions:** Conceptualization, B.K. Methodology, B.K.; Software, Not applicable; Validation, B.K.; Formal Analysis, B.K..; Investigation, B.K., P.P.; Resources, B.K.; Data Curation, B.K., P.P.; Writing-Original Draft Preparation, B.K., P.P.; Writing-Review & Editing, B.K.; Visualization, B.K.; Supervision, B.K.; Project Administration, N/A; Funding Acquisition, N/A. All authors have read and agreed to the published version of the manuscript.

**Funding:** This research received no external funding.

**Acknowledgments:** This paper was supported by the research project SPEV 2020, run at the Faculty of Informatics and Management, University of Hradec Kralove, Czech Republic. The authors thank Aleš Berger for his help with data collection.

**Conflicts of Interest:** The authors declare no conflict of interest.

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
