# Peer review of "Students’ Perceptions of an EFL Vocabulary Learning Mobile Application"

_education, doi:10.3390/educsci10020037_

Round 1
Reviewer 1 Report
I add my comments in the attached .pdf document

Reviewer 2 Report
The topic is in line with modern tendences (blended learning, mobile apps, students` needs, etc.). The article is well-structured and written in good English, though the author should check a couple of lines (103,104, 150,234). for grammar errors. A wide range of relevant findings, referances and statistics make the author`s arguments clear and well-grounded.
The author gives a detailed review of the advantages of smartphones for learning, but there are a few repetitions of the same ideas (for example, lines 93-94/11-12). Later, in the discussion part the author again mentions some of them. I would strongly advise the author to reorganize this part (literature review) of the article, adding a couple of points which focus on the disadvantages of mobile apps as well. There are no ideal aids in teaching and learning.
The description of the study arouses a lot of questions: which words were checked in the pre-test? Definitely some different ones. Therefore, Table 2 looks irrelevant. Did the assignments on the app consist only of translation? That`s what I got from the text of the article. The final post-test also required translation. Does it mean that translation exercises were the only type of activity that the students fulfilled? If it is so, then we return to the translation method of FL teaching and learning used 100 years ago. It seems that the focus of the study was to help the students remember and translate the words. I would recommend the author to give more details of the tasks that the students had to do. What makes the author believe that the students` progress was due exceptionally to the apps, but not to the tasks on the Blackboard or to their face-to-face practice? Besides, the author mustn`t lose focus in the descriptive part providing a more profound analysis of the results of the questionnaire. Unfortunately, many of them are left out.
It is evident that the article needs to be improved before being published it in the journal.
Round 2
Reviewer 1 Report
Manuscript title: Students’ perceptions of an EFL Vocabulary Learning Mobile Application
2nd review
The authors have addressed most of the concerns raised in the previous review. I have further comments displayed below. The article needs another revision round, in my opinion. It is not ready to publish yet.
Abstract:
Lines 6-7: Especially smartphones are becoming…à Previous research has shown that smartphones are becoming widely used in learning as they can improve knowledge retention and increase student engagement.
Lines 8-10: The purpose of this study is to discuss students’ perception of the use of a mobile application, which was developed on the basis of students needs, and describe its strengths and weaknesses as perceived by the students à The purpose of this study is to discuss students’ perception of the use of a mobile application aimed at learning new English vocabulary and phrases and describe its strengths and weaknesses as perceived by the students.
Line 10: This mobile application is aimed… à eliminate sentence.
Line 12: the results showed that the positive aspects of the mobile app prevailed over the negative ones àOverall, the students agreement to the positive aspects displayed in the questionnaire prevailed over their disagreement or neutral opinions.
Line 15: Develop more the more items which displayed more disagreement, here. They also need to be addressed and are even more interesting for the research community.
Introduction:
Line 23: respectively, smartphonesà eliminate (smartphones and phones are not distinguishable in this community; they ALL use smartphones, to my view)
Line 44: In addition à as a consequence,
Line 45: …as it has been evinced in research studies à as has been shown
Line 46: .. studying English language preparatory course à studying an English language preparatory course
Line 57: …by a mobile system., à …by a mobile device,
Lines 49-60: This looks much neater now with the addition of some studies which evince the claim stated in line 45.
Line 59: …and the practice of speaking EFL in an authentic context. à this is not an measurable outcome, as you are displaying before. Eliminate or rephrase
Line 61: Explain ‘pure’
Lines 65: no need for new paragraph
lines 68-71: the way Schmidt has been introduced is oversimplified and secondary. As the text stands, the authors do not need to justify why learning vocabulary in context is important, they are just relating the statistics of vocabulary apps. Also, their app does not cover it.
If the authors wish to develop some lines to the importance of learning vocabulary in context, this must be developed more thoroughly. I do not think this is necessary in this manuscript.
Line 77: In accordance with the statement above, Mikulecky [32] writes that …. à Authors such as Mikulecky [32] also claim that …
No need for new paragraph.
Lines 82-86: According to him a valuable [….] face-to-face learning methods. à Eliminate
Repeated ideas
Line 98-101: I do not see the connection between having access to ‘information all over the world’ and fixed nature and teacher provision of information. What is wrong with teacher information? What is wrong with fixed schedules? Are they not also part of blended learning? Is the ‘information all over the world’ more accurate than the one provided by the teacher? Is it that ‘fixed’ timetables should be eliminated altogether? rewrite
Line: 105: explain ‘noisy environment’
Method
Line 133: a method of a pen and paper questionnaire survey was used…à A pen and paper questionnaire was completed by 28 students of Management of tourism in their third academic year. It aimed at exploring students’ perceptions about the use of a mobile app which they were using during the winter semester of 2019 for learning new words and phrases.
Lines 138-140: eliminate last sentence of paragraph
Line 151: explain the on-line blackboard. How does Figure 1 support the information about the on-line blackboard? I am not sure it is informatively relevant. Also, the quality of the image is poor. Please consider the need to keep or otherwise eliminate figure.
Figure 2: The quality of the image is poor. Please consider the need to keep or otherwise eliminate figure.
Figure 3: The quality of the image is poor. I am not sure it is informatively relevant. Please consider the need to keep or otherwise eliminate figure.
Results and discussion
Lines 179-181: Add gender and origin of informants in Methods section.
Line 182: The replies to those two questions are gen discussed below. à The replies to the two open questions are discussed below.
Lines 182-183: The authors counted the agree and strongly agree answers as positive, while the strongly disagree and disagree as negative.à The authors interpreted the ‘agree’ and ‘strongly agree’ responses as students positively valuing the learning statement proposed in the item questionnaire. The ‘strongly disagree’ and ‘disagree’ responses were interpreted as students valuing the statements as negative. Those items in which positive answers sum more than 51% are interpreted.
Line 191: positive à positive, as more than 51% of the responses were rated as ‘agree’ or strongly agree’.
Line 196: This was confirmed by the findings of other research studies… à This goes in line with the findings of other research studies…
Figure 4: The quality of the image is poor. I am not sure it is informatively relevant. Please consider the need to keep or otherwise eliminate figure.
Lines 198-209: The students appreciated the provision of corrective feedback by the app. The authors have interpreted this as synonym of formative assessment. I am not sure these two aspects are to be interpreted as such. Please clarify
Line 211: , which is slightly surprising since only four answers were positive.
The data do show that the students found some benefits in the app. The thing is that you do not necessarily have to find ALL the benefits addressed in the questionnaire. The few very positive reports are enough for the students to state that they would like to use it again. If it worked for some things, why not use it again, fair enough. Eliminate the sentence.
Lines 2015-228: the authors have also interpreted those statements which were rated less positively (below 51%). This is good. However, I wonder whether the blended learning (mobile + contact classes) may be interacting here. I mean, one ‘charm’ of a combined teaching approach such as blended learning might be that it appeals to the different learning styles of the students. Could it be that those students who are not reporting that the app helped are just enjoying/benefiting the more traditional side of the blended procedure: the contact clases?
Line 229: On the contrary à eliminate
Line 230: ... considered it to be very strict
where does these data come from? Open questions? Specify
Line 230: the lack of dot à the lack of a dot.
Do you mean full stop?
Line 232: which was not used by majority of students à which was not used by the majority of the students
Line 233: explain what is the app’s interface as for the pronunciation of the new words. An audio, a phonetic transcription? Was it an optional action? A B2 level can well still feel very insecure about pronunciation, specially of new words. This argument may not be the only one the authors should contemplate here. Was the oral skill (and therefore pronunciation) ever assessed? How was assessment proceduralised? Orally? Written? Translation? These may have had an impact on how the students approached their vocabulary learning.
Line 241: Nevertheless, the authors of this study are persuaded that language learners must master vocabulary first in order to be able to express themselves […] help in this respect. à However, bottom-up vocabulary processing can also be an optimal learning strategy, specially if it is associated to a later stage in which the learner can use it in context, as it is the case of the presented blended learning procedure here.
Line 245-247: Could it be that the T notifications were not learning-supportive, or tailored? As they look in Figure 4, they are mere informative/reminder notes, which could have been generated by the system itself. Maybe the format/content of the notifications was not productive/significant for the student??
Conclusions
Lines 252: may perceive à perceived
Line 254: add all the positive aspects here. Highlight your findings more.
Line 254: On the other hand, as the findings indicate, more attention should be paid…à On the other hand, as the findings indicate, the students reported that the app was not very supportive as for communication performance; they did not find the teachers’ notifications encouraging and they did not use the pronunciation support much.
I think that these three aspects along with the notorious neutral responses still need to be accounted for in this manuscript.
Manuscript title: Students’ perceptions of an EFL Vocabulary Learning Mobile Application
2nd review
The authors have addressed most of the concerns raised in the previous review. I have further comments displayed below. The article needs another revision round, in my opinion. It is not ready to publish yet.
Abstract:
Lines 6-7: Especially smartphones are becoming…à Previous research has shown that smartphones are becoming widely used in learning as they can improve knowledge retention and increase student engagement.
Lines 8-10: The purpose of this study is to discuss students’ perception of the use of a mobile application, which was developed on the basis of students needs, and describe its strengths and weaknesses as perceived by the students à The purpose of this study is to discuss students’ perception of the use of a mobile application aimed at learning new English vocabulary and phrases and describe its strengths and weaknesses as perceived by the students.
Line 10: This mobile application is aimed… à eliminate sentence.
Line 12: the results showed that the positive aspects of the mobile app prevailed over the negative ones àOverall, the students agreement to the positive aspects displayed in the questionnaire prevailed over their disagreement or neutral opinions.
Line 15: Develop more the more items which displayed more disagreement, here. They also need to be addressed and are even more interesting for the research community.
Introduction:
Line 23: respectively, smartphonesà eliminate (smartphones and phones are not distinguishable in this community; they ALL use smartphones, to my view)
Line 44: In addition à as a consequence,
Line 45: …as it has been evinced in research studies à as has been shown
Line 46: .. studying English language preparatory course à studying an English language preparatory course
Line 57: …by a mobile system., à …by a mobile device,
Lines 49-60: This looks much neater now with the addition of some studies which evince the claim stated in line 45.
Line 59: …and the practice of speaking EFL in an authentic context. à this is not an measurable outcome, as you are displaying before. Eliminate or rephrase
Line 61: Explain ‘pure’
Lines 65: no need for new paragraph
lines 68-71: the way Schmidt has been introduced is oversimplified and secondary. As the text stands, the authors do not need to justify why learning vocabulary in context is important, they are just relating the statistics of vocabulary apps. Also, their app does not cover it.
If the authors wish to develop some lines to the importance of learning vocabulary in context, this must be developed more thoroughly. I do not think this is necessary in this manuscript.
Line 77: In accordance with the statement above, Mikulecky [32] writes that …. à Authors such as Mikulecky [32] also claim that …
No need for new paragraph.
Lines 82-86: According to him a valuable [….] face-to-face learning methods. à Eliminate
Repeated ideas
Line 98-101: I do not see the connection between having access to ‘information all over the world’ and fixed nature and teacher provision of information. What is wrong with teacher information? What is wrong with fixed schedules? Are they not also part of blended learning? Is the ‘information all over the world’ more accurate than the one provided by the teacher? Is it that ‘fixed’ timetables should be eliminated altogether? rewrite
Line: 105: explain ‘noisy environment’
Method
Line 133: a method of a pen and paper questionnaire survey was used…à A pen and paper questionnaire was completed by 28 students of Management of tourism in their third academic year. It aimed at exploring students’ perceptions about the use of a mobile app which they were using during the winter semester of 2019 for learning new words and phrases.
Lines 138-140: eliminate last sentence of paragraph
Line 151: explain the on-line blackboard. How does Figure 1 support the information about the on-line blackboard? I am not sure it is informatively relevant. Also, the quality of the image is poor. Please consider the need to keep or otherwise eliminate figure.
Figure 2: The quality of the image is poor. Please consider the need to keep or otherwise eliminate figure.
Figure 3: The quality of the image is poor. I am not sure it is informatively relevant. Please consider the need to keep or otherwise eliminate figure.
Results and discussion
Lines 179-181: Add gender and origin of informants in Methods section.
Line 182: The replies to those two questions are gen discussed below. à The replies to the two open questions are discussed below.
Lines 182-183: The authors counted the agree and strongly agree answers as positive, while the strongly disagree and disagree as negative.à The authors interpreted the ‘agree’ and ‘strongly agree’ responses as students positively valuing the learning statement proposed in the item questionnaire. The ‘strongly disagree’ and ‘disagree’ responses were interpreted as students valuing the statements as negative. Those items in which positive answers sum more than 51% are interpreted.
Line 191: positive à positive, as more than 51% of the responses were rated as ‘agree’ or strongly agree’.
Line 196: This was confirmed by the findings of other research studies… à This goes in line with the findings of other research studies…
Figure 4: The quality of the image is poor. I am not sure it is informatively relevant. Please consider the need to keep or otherwise eliminate figure.
Lines 198-209: The students appreciated the provision of corrective feedback by the app. The authors have interpreted this as synonym of formative assessment. I am not sure these two aspects are to be interpreted as such. Please clarify
Line 211: , which is slightly surprising since only four answers were positive.
The data do show that the students found some benefits in the app. The thing is that you do not necessarily have to find ALL the benefits addressed in the questionnaire. The few very positive reports are enough for the students to state that they would like to use it again. If it worked for some things, why not use it again, fair enough. Eliminate the sentence.
Lines 2015-228: the authors have also interpreted those statements which were rated less positively (below 51%). This is good. However, I wonder whether the blended learning (mobile + contact classes) may be interacting here. I mean, one ‘charm’ of a combined teaching approach such as blended learning might be that it appeals to the different learning styles of the students. Could it be that those students who are not reporting that the app helped are just enjoying/benefiting the more traditional side of the blended procedure: the contact clases?
Line 229: On the contrary à eliminate
Line 230: ... considered it to be very strict
where does these data come from? Open questions? Specify
Line 230: the lack of dot à the lack of a dot.
Do you mean full stop?
Line 232: which was not used by majority of students à which was not used by the majority of the students
Line 233: explain what is the app’s interface as for the pronunciation of the new words. An audio, a phonetic transcription? Was it an optional action? A B2 level can well still feel very insecure about pronunciation, specially of new words. This argument may not be the only one the authors should contemplate here. Was the oral skill (and therefore pronunciation) ever assessed? How was assessment proceduralised? Orally? Written? Translation? These may have had an impact on how the students approached their vocabulary learning.
Line 241: Nevertheless, the authors of this study are persuaded that language learners must master vocabulary first in order to be able to express themselves […] help in this respect. à However, bottom-up vocabulary processing can also be an optimal learning strategy, specially if it is associated to a later stage in which the learner can use it in context, as it is the case of the presented blended learning procedure here.
Line 245-247: Could it be that the T notifications were not learning-supportive, or tailored? As they look in Figure 4, they are mere informative/reminder notes, which could have been generated by the system itself. Maybe the format/content of the notifications was not productive/significant for the student??
Conclusions
Lines 252: may perceive à perceived
Line 254: add all the positive aspects here. Highlight your findings more.
Line 254: On the other hand, as the findings indicate, more attention should be paid…à On the other hand, as the findings indicate, the students reported that the app was not very supportive as for communication performance; they did not find the teachers’ notifications encouraging and they did not use the pronunciation support much.
I think that these three aspects along with the notorious neutral responses still need to be accounted for in this manuscript.
Manuscript title: Students’ perceptions of an EFL Vocabulary Learning Mobile Application
2nd review
The authors have addressed most of the concerns raised in the previous review. I have further comments displayed below. The article needs another revision round, in my opinion. It is not ready to publish yet.
Abstract:
Lines 6-7: Especially smartphones are becoming…à Previous research has shown that smartphones are becoming widely used in learning as they can improve knowledge retention and increase student engagement.
Lines 8-10: The purpose of this study is to discuss students’ perception of the use of a mobile application, which was developed on the basis of students needs, and describe its strengths and weaknesses as perceived by the students à The purpose of this study is to discuss students’ perception of the use of a mobile application aimed at learning new English vocabulary and phrases and describe its strengths and weaknesses as perceived by the students.
Line 10: This mobile application is aimed… à eliminate sentence.
Line 12: the results showed that the positive aspects of the mobile app prevailed over the negative ones àOverall, the students agreement to the positive aspects displayed in the questionnaire prevailed over their disagreement or neutral opinions.
Line 15: Develop more the more items which displayed more disagreement, here. They also need to be addressed and are even more interesting for the research community.
Introduction:
Line 23: respectively, smartphonesà eliminate (smartphones and phones are not distinguishable in this community; they ALL use smartphones, to my view)
Line 44: In addition à as a consequence,
Line 45: …as it has been evinced in research studies à as has been shown
Line 46: .. studying English language preparatory course à studying an English language preparatory course
Line 57: …by a mobile system., à …by a mobile device,
Lines 49-60: This looks much neater now with the addition of some studies which evince the claim stated in line 45.
Line 59: …and the practice of speaking EFL in an authentic context. à this is not an measurable outcome, as you are displaying before. Eliminate or rephrase
Line 61: Explain ‘pure’
Lines 65: no need for new paragraph
lines 68-71: the way Schmidt has been introduced is oversimplified and secondary. As the text stands, the authors do not need to justify why learning vocabulary in context is important, they are just relating the statistics of vocabulary apps. Also, their app does not cover it.
If the authors wish to develop some lines to the importance of learning vocabulary in context, this must be developed more thoroughly. I do not think this is necessary in this manuscript.
Line 77: In accordance with the statement above, Mikulecky [32] writes that …. à Authors such as Mikulecky [32] also claim that …
No need for new paragraph.
Lines 82-86: According to him a valuable [….] face-to-face learning methods. à Eliminate
Repeated ideas
Line 98-101: I do not see the connection between having access to ‘information all over the world’ and fixed nature and teacher provision of information. What is wrong with teacher information? What is wrong with fixed schedules? Are they not also part of blended learning? Is the ‘information all over the world’ more accurate than the one provided by the teacher? Is it that ‘fixed’ timetables should be eliminated altogether? rewrite
Line: 105: explain ‘noisy environment’
Method
Line 133: a method of a pen and paper questionnaire survey was used…à A pen and paper questionnaire was completed by 28 students of Management of tourism in their third academic year. It aimed at exploring students’ perceptions about the use of a mobile app which they were using during the winter semester of 2019 for learning new words and phrases.
Lines 138-140: eliminate last sentence of paragraph
Line 151: explain the on-line blackboard. How does Figure 1 support the information about the on-line blackboard? I am not sure it is informatively relevant. Also, the quality of the image is poor. Please consider the need to keep or otherwise eliminate figure.
Figure 2: The quality of the image is poor. Please consider the need to keep or otherwise eliminate figure.
Figure 3: The quality of the image is poor. I am not sure it is informatively relevant. Please consider the need to keep or otherwise eliminate figure.
Results and discussion
Lines 179-181: Add gender and origin of informants in Methods section.
Line 182: The replies to those two questions are gen discussed below. à The replies to the two open questions are discussed below.
Lines 182-183: The authors counted the agree and strongly agree answers as positive, while the strongly disagree and disagree as negative.à The authors interpreted the ‘agree’ and ‘strongly agree’ responses as students positively valuing the learning statement proposed in the item questionnaire. The ‘strongly disagree’ and ‘disagree’ responses were interpreted as students valuing the statements as negative. Those items in which positive answers sum more than 51% are interpreted.
Line 191: positive à positive, as more than 51% of the responses were rated as ‘agree’ or strongly agree’.
Line 196: This was confirmed by the findings of other research studies… à This goes in line with the findings of other research studies…
Figure 4: The quality of the image is poor. I am not sure it is informatively relevant. Please consider the need to keep or otherwise eliminate figure.
Lines 198-209: The students appreciated the provision of corrective feedback by the app. The authors have interpreted this as synonym of formative assessment. I am not sure these two aspects are to be interpreted as such. Please clarify
Line 211: , which is slightly surprising since only four answers were positive.
The data do show that the students found some benefits in the app. The thing is that you do not necessarily have to find ALL the benefits addressed in the questionnaire. The few very positive reports are enough for the students to state that they would like to use it again. If it worked for some things, why not use it again, fair enough. Eliminate the sentence.
Lines 2015-228: the authors have also interpreted those statements which were rated less positively (below 51%). This is good. However, I wonder whether the blended learning (mobile + contact classes) may be interacting here. I mean, one ‘charm’ of a combined teaching approach such as blended learning might be that it appeals to the different learning styles of the students. Could it be that those students who are not reporting that the app helped are just enjoying/benefiting the more traditional side of the blended procedure: the contact clases?
Line 229: On the contrary à eliminate
Line 230: ... considered it to be very strict
where does these data come from? Open questions? Specify
Line 230: the lack of dot à the lack of a dot.
Do you mean full stop?
Line 232: which was not used by majority of students à which was not used by the majority of the students
Line 233: explain what is the app’s interface as for the pronunciation of the new words. An audio, a phonetic transcription? Was it an optional action? A B2 level can well still feel very insecure about pronunciation, specially of new words. This argument may not be the only one the authors should contemplate here. Was the oral skill (and therefore pronunciation) ever assessed? How was assessment proceduralised? Orally? Written? Translation? These may have had an impact on how the students approached their vocabulary learning.
Line 241: Nevertheless, the authors of this study are persuaded that language learners must master vocabulary first in order to be able to express themselves […] help in this respect. à However, bottom-up vocabulary processing can also be an optimal learning strategy, specially if it is associated to a later stage in which the learner can use it in context, as it is the case of the presented blended learning procedure here.
Line 245-247: Could it be that the T notifications were not learning-supportive, or tailored? As they look in Figure 4, they are mere informative/reminder notes, which could have been generated by the system itself. Maybe the format/content of the notifications was not productive/significant for the student??
Conclusions
Lines 252: may perceive à perceived
Line 254: add all the positive aspects here. Highlight your findings more.
Line 254: On the other hand, as the findings indicate, more attention should be paid…à On the other hand, as the findings indicate, the students reported that the app was not very supportive as for communication performance; they did not find the teachers’ notifications encouraging and they did not use the pronunciation support much.
I think that these three aspects along with the notorious neutral responses still need to be accounted for in this manuscript.
Manuscript title: Students’ perceptions of an EFL Vocabulary Learning Mobile Application
2nd review
The authors have addressed most of the concerns raised in the previous review. I have further comments displayed below. The article needs another revision round, in my opinion. It is not ready to publish yet.
Abstract:
Lines 6-7: Especially smartphones are becoming…à Previous research has shown that smartphones are becoming widely used in learning as they can improve knowledge retention and increase student engagement.
Lines 8-10: The purpose of this study is to discuss students’ perception of the use of a mobile application, which was developed on the basis of students needs, and describe its strengths and weaknesses as perceived by the students à The purpose of this study is to discuss students’ perception of the use of a mobile application aimed at learning new English vocabulary and phrases and describe its strengths and weaknesses as perceived by the students.
Line 10: This mobile application is aimed… à eliminate sentence.
Line 12: the results showed that the positive aspects of the mobile app prevailed over the negative ones àOverall, the students agreement to the positive aspects displayed in the questionnaire prevailed over their disagreement or neutral opinions.
Line 15: Develop more the more items which displayed more disagreement, here. They also need to be addressed and are even more interesting for the research community.
Introduction:
Line 23: respectively, smartphonesà eliminate (smartphones and phones are not distinguishable in this community; they ALL use smartphones, to my view)
Line 44: In addition à as a consequence,
Line 45: …as it has been evinced in research studies à as has been shown
Line 46: .. studying English language preparatory course à studying an English language preparatory course
Line 57: …by a mobile system., à …by a mobile device,
Lines 49-60: This looks much neater now with the addition of some studies which evince the claim stated in line 45.
Line 59: …and the practice of speaking EFL in an authentic context. à this is not an measurable outcome, as you are displaying before. Eliminate or rephrase
Line 61: Explain ‘pure’
Lines 65: no need for new paragraph
lines 68-71: the way Schmidt has been introduced is oversimplified and secondary. As the text stands, the authors do not need to justify why learning vocabulary in context is important, they are just relating the statistics of vocabulary apps. Also, their app does not cover it.
If the authors wish to develop some lines to the importance of learning vocabulary in context, this must be developed more thoroughly. I do not think this is necessary in this manuscript.
Line 77: In accordance with the statement above, Mikulecky [32] writes that …. à Authors such as Mikulecky [32] also claim that …
No need for new paragraph.
Lines 82-86: According to him a valuable [….] face-to-face learning methods. à Eliminate
Repeated ideas
Line 98-101: I do not see the connection between having access to ‘information all over the world’ and fixed nature and teacher provision of information. What is wrong with teacher information? What is wrong with fixed schedules? Are they not also part of blended learning? Is the ‘information all over the world’ more accurate than the one provided by the teacher? Is it that ‘fixed’ timetables should be eliminated altogether? rewrite
Line: 105: explain ‘noisy environment’
Method
Line 133: a method of a pen and paper questionnaire survey was used…à A pen and paper questionnaire was completed by 28 students of Management of tourism in their third academic year. It aimed at exploring students’ perceptions about the use of a mobile app which they were using during the winter semester of 2019 for learning new words and phrases.
Lines 138-140: eliminate last sentence of paragraph
Line 151: explain the on-line blackboard. How does Figure 1 support the information about the on-line blackboard? I am not sure it is informatively relevant. Also, the quality of the image is poor. Please consider the need to keep or otherwise eliminate figure.
Figure 2: The quality of the image is poor. Please consider the need to keep or otherwise eliminate figure.
Figure 3: The quality of the image is poor. I am not sure it is informatively relevant. Please consider the need to keep or otherwise eliminate figure.
Results and discussion
Lines 179-181: Add gender and origin of informants in Methods section.
Line 182: The replies to those two questions are gen discussed below. à The replies to the two open questions are discussed below.
Lines 182-183: The authors counted the agree and strongly agree answers as positive, while the strongly disagree and disagree as negative.à The authors interpreted the ‘agree’ and ‘strongly agree’ responses as students positively valuing the learning statement proposed in the item questionnaire. The ‘strongly disagree’ and ‘disagree’ responses were interpreted as students valuing the statements as negative. Those items in which positive answers sum more than 51% are interpreted.
Line 191: positive à positive, as more than 51% of the responses were rated as ‘agree’ or strongly agree’.
Line 196: This was confirmed by the findings of other research studies… à This goes in line with the findings of other research studies…
Figure 4: The quality of the image is poor. I am not sure it is informatively relevant. Please consider the need to keep or otherwise eliminate figure.
Lines 198-209: The students appreciated the provision of corrective feedback by the app. The authors have interpreted this as synonym of formative assessment. I am not sure these two aspects are to be interpreted as such. Please clarify
Line 211: , which is slightly surprising since only four answers were positive.
The data do show that the students found some benefits in the app. The thing is that you do not necessarily have to find ALL the benefits addressed in the questionnaire. The few very positive reports are enough for the students to state that they would like to use it again. If it worked for some things, why not use it again, fair enough. Eliminate the sentence.
Lines 2015-228: the authors have also interpreted those statements which were rated less positively (below 51%). This is good. However, I wonder whether the blended learning (mobile + contact classes) may be interacting here. I mean, one ‘charm’ of a combined teaching approach such as blended learning might be that it appeals to the different learning styles of the students. Could it be that those students who are not reporting that the app helped are just enjoying/benefiting the more traditional side of the blended procedure: the contact clases?
Line 229: On the contrary à eliminate
Line 230: ... considered it to be very strict
where does these data come from? Open questions? Specify
Line 230: the lack of dot à the lack of a dot.
Do you mean full stop?
Line 232: which was not used by majority of students à which was not used by the majority of the students
Line 233: explain what is the app’s interface as for the pronunciation of the new words. An audio, a phonetic transcription? Was it an optional action? A B2 level can well still feel very insecure about pronunciation, specially of new words. This argument may not be the only one the authors should contemplate here. Was the oral skill (and therefore pronunciation) ever assessed? How was assessment proceduralised? Orally? Written? Translation? These may have had an impact on how the students approached their vocabulary learning.
Line 241: Nevertheless, the authors of this study are persuaded that language learners must master vocabulary first in order to be able to express themselves […] help in this respect. à However, bottom-up vocabulary processing can also be an optimal learning strategy, specially if it is associated to a later stage in which the learner can use it in context, as it is the case of the presented blended learning procedure here.
Line 245-247: Could it be that the T notifications were not learning-supportive, or tailored? As they look in Figure 4, they are mere informative/reminder notes, which could have been generated by the system itself. Maybe the format/content of the notifications was not productive/significant for the student??
Conclusions
Lines 252: may perceive à perceived
Line 254: add all the positive aspects here. Highlight your findings more.
Line 254: On the other hand, as the findings indicate, more attention should be paid…à On the other hand, as the findings indicate, the students reported that the app was not very supportive as for communication performance; they did not find the teachers’ notifications encouraging and they did not use the pronunciation support much.
I think that these three aspects along with the notorious neutral responses still need to be accounted for in this manuscript.
Manuscript title: Students’ perceptions of an EFL Vocabulary Learning Mobile Application
2nd review
The authors have addressed most of the concerns raised in the previous review. I have further comments displayed below. The article needs another revision round, in my opinion. It is not ready to publish yet.
Abstract:
Lines 6-7: Especially smartphones are becoming…à Previous research has shown that smartphones are becoming widely used in learning as they can improve knowledge retention and increase student engagement.
Lines 8-10: The purpose of this study is to discuss students’ perception of the use of a mobile application, which was developed on the basis of students needs, and describe its strengths and weaknesses as perceived by the students à The purpose of this study is to discuss students’ perception of the use of a mobile application aimed at learning new English vocabulary and phrases and describe its strengths and weaknesses as perceived by the students.
Line 10: This mobile application is aimed… à eliminate sentence.
Line 12: the results showed that the positive aspects of the mobile app prevailed over the negative ones àOverall, the students agreement to the positive aspects displayed in the questionnaire prevailed over their disagreement or neutral opinions.
Line 15: Develop more the more items which displayed more disagreement, here. They also need to be addressed and are even more interesting for the research community.
Introduction:
Line 23: respectively, smartphonesà eliminate (smartphones and phones are not distinguishable in this community; they ALL use smartphones, to my view)
Line 44: In addition à as a consequence,
Line 45: …as it has been evinced in research studies à as has been shown
Line 46: .. studying English language preparatory course à studying an English language preparatory course
Line 57: …by a mobile system., à …by a mobile device,
Lines 49-60: This looks much neater now with the addition of some studies which evince the claim stated in line 45.
Line 59: …and the practice of speaking EFL in an authentic context. à this is not an measurable outcome, as you are displaying before. Eliminate or rephrase
Line 61: Explain ‘pure’
Lines 65: no need for new paragraph
lines 68-71: the way Schmidt has been introduced is oversimplified and secondary. As the text stands, the authors do not need to justify why learning vocabulary in context is important, they are just relating the statistics of vocabulary apps. Also, their app does not cover it.
If the authors wish to develop some lines to the importance of learning vocabulary in context, this must be developed more thoroughly. I do not think this is necessary in this manuscript.
Line 77: In accordance with the statement above, Mikulecky [32] writes that …. à Authors such as Mikulecky [32] also claim that …
No need for new paragraph.
Lines 82-86: According to him a valuable [….] face-to-face learning methods. à Eliminate
Repeated ideas
Line 98-101: I do not see the connection between having access to ‘information all over the world’ and fixed nature and teacher provision of information. What is wrong with teacher information? What is wrong with fixed schedules? Are they not also part of blended learning? Is the ‘information all over the world’ more accurate than the one provided by the teacher? Is it that ‘fixed’ timetables should be eliminated altogether? rewrite
Line: 105: explain ‘noisy environment’
Method
Line 133: a method of a pen and paper questionnaire survey was used…à A pen and paper questionnaire was completed by 28 students of Management of tourism in their third academic year. It aimed at exploring students’ perceptions about the use of a mobile app which they were using during the winter semester of 2019 for learning new words and phrases.
Lines 138-140: eliminate last sentence of paragraph
Line 151: explain the on-line blackboard. How does Figure 1 support the information about the on-line blackboard? I am not sure it is informatively relevant. Also, the quality of the image is poor. Please consider the need to keep or otherwise eliminate figure.
Figure 2: The quality of the image is poor. Please consider the need to keep or otherwise eliminate figure.
Figure 3: The quality of the image is poor. I am not sure it is informatively relevant. Please consider the need to keep or otherwise eliminate figure.
Results and discussion
Lines 179-181: Add gender and origin of informants in Methods section.
Line 182: The replies to those two questions are gen discussed below. à The replies to the two open questions are discussed below.
Lines 182-183: The authors counted the agree and strongly agree answers as positive, while the strongly disagree and disagree as negative.à The authors interpreted the ‘agree’ and ‘strongly agree’ responses as students positively valuing the learning statement proposed in the item questionnaire. The ‘strongly disagree’ and ‘disagree’ responses were interpreted as students valuing the statements as negative. Those items in which positive answers sum more than 51% are interpreted.
Line 191: positive à positive, as more than 51% of the responses were rated as ‘agree’ or strongly agree’.
Line 196: This was confirmed by the findings of other research studies… à This goes in line with the findings of other research studies…
Figure 4: The quality of the image is poor. I am not sure it is informatively relevant. Please consider the need to keep or otherwise eliminate figure.
Lines 198-209: The students appreciated the provision of corrective feedback by the app. The authors have interpreted this as synonym of formative assessment. I am not sure these two aspects are to be interpreted as such. Please clarify
Line 211: , which is slightly surprising since only four answers were positive.
The data do show that the students found some benefits in the app. The thing is that you do not necessarily have to find ALL the benefits addressed in the questionnaire. The few very positive reports are enough for the students to state that they would like to use it again. If it worked for some things, why not use it again, fair enough. Eliminate the sentence.
Lines 2015-228: the authors have also interpreted those statements which were rated less positively (below 51%). This is good. However, I wonder whether the blended learning (mobile + contact classes) may be interacting here. I mean, one ‘charm’ of a combined teaching approach such as blended learning might be that it appeals to the different learning styles of the students. Could it be that those students who are not reporting that the app helped are just enjoying/benefiting the more traditional side of the blended procedure: the contact clases?
Line 229: On the contrary à eliminate
Line 230: ... considered it to be very strict
where does these data come from? Open questions? Specify
Line 230: the lack of dot à the lack of a dot.
Do you mean full stop?
Line 232: which was not used by majority of students à which was not used by the majority of the students
Line 233: explain what is the app’s interface as for the pronunciation of the new words. An audio, a phonetic transcription? Was it an optional action? A B2 level can well still feel very insecure about pronunciation, specially of new words. This argument may not be the only one the authors should contemplate here. Was the oral skill (and therefore pronunciation) ever assessed? How was assessment proceduralised? Orally? Written? Translation? These may have had an impact on how the students approached their vocabulary learning.
Line 241: Nevertheless, the authors of this study are persuaded that language learners must master vocabulary first in order to be able to express themselves […] help in this respect. à However, bottom-up vocabulary processing can also be an optimal learning strategy, specially if it is associated to a later stage in which the learner can use it in context, as it is the case of the presented blended learning procedure here.
Line 245-247: Could it be that the T notifications were not learning-supportive, or tailored? As they look in Figure 4, they are mere informative/reminder notes, which could have been generated by the system itself. Maybe the format/content of the notifications was not productive/significant for the student??
Conclusions
Lines 252: may perceive à perceived
Line 254: add all the positive aspects here. Highlight your findings more.
Line 254: On the other hand, as the findings indicate, more attention should be paid…à On the other hand, as the findings indicate, the students reported that the app was not very supportive as for communication performance; they did not find the teachers’ notifications encouraging and they did not use the pronunciation support much.
I think that these three aspects along with the notorious neutral responses still need to be accounted for in this manuscript.
Reviewer 2 Report
The authors have done a lot of improvements and tried to explain their position having answered the questions in my previous review. Their have also provided sufficient comments concerning the analysis of the research results. The article gives a clear view of what was happening, what the students were supposed to do and how they were learning the vocabulary. Though I do not agree with the authors` opinion about translation assignments as a primary way of vocabulary acquisition, I admit that there may be a different approach.
My suggestion is that after the authors have corrected a few errors (lines 76, 97,230) and checked the text for the use of articles, the article can be published.
